

# Detection and visualization of abnormality in chest radiographs using modality-specific convolutional neural network ensembles

Sivaramakrishnan Rajaraman, Incheol Kim and Sameer K. Antani

Lister Hill National Center for Biomedical Communications, National Library of Medicine, National Institutes of Health, Bethesda, MD, United States of America

## ABSTRACT

Convolutional neural networks (CNNs) trained on natural images are extremely successful in image classification and localization due to superior automated feature extraction capability. In extending their use to biomedical recognition tasks, it is important to note that visual features of medical images tend to be uniquely different than natural images. There are advantages offered through training these networks on large scale medical common modality image collections pertaining to the recognition task. Further, improved generalization in transferring knowledge across similar tasks is possible when the models are trained to learn modality-specific features and then suitably repurposed for the target task. In this study, we propose modality-specific ensemble learning toward improving abnormality detection in chest X-rays (CXRs). CNN models are trained on a large-scale CXR collection to learn modality-specific features and then repurposed for detecting and localizing abnormalities. Model predictions are combined using different ensemble strategies toward reducing prediction variance and sensitivity to the training data while improving overall performance and generalization. Class-selective relevance mapping (CRM) is used to visualize the learned behavior of the individual models and their ensembles. It localizes discriminative regions of interest (ROIs) showing abnormal regions and offers an improved explanation of model predictions. It was observed that the model ensembles demonstrate superior localization performance in terms of Intersection of Union (IoU) and mean Average Precision (mAP) metrics than any individual constituent model.

# INTRODUCTION

Computer-aided diagnosis (CADx) tools have gained immense prominence in medicine by augmenting clinical expertise and reducing observer variability (*Bar et al., 2015*). Data-driven deep learning (DL) algorithms using convolutional neural networks (CNNs) have been successfully applied to chest X-ray (CXR) screening (*Singh et al., 2018*; *Rajpurkar et al., 2018*; *Qin et al., 2018*; *Irvin et al., 2019*; *Pasa et al., 2019*). The CXRs are analyzed for typical abnormalities to localize suspicious regions (*Hwang et al., 2016*;

Corresponding author
Sivaramakrishnan Rajaraman,
sivaramakrishnan.rajaraman@nih.gov,
raaju.shiv1@gmail.com

*Lakhani & Sundaram, 2017*; *Xiong et al., 2018*). However, the models are disease-specific. Generalization would require retraining models on additional expert-annotated training data and labels. This can be time-consuming and expensive.

Transfer Learning (TF) strategies are commonly adopted when limited data is available, e.g., medical images. Here, CNNs are trained on a large-scale selection of natural images and the learned knowledge is transferred and repurposed for the new task (*Lopes & Valiati, 2017*; *Rajaraman et al., 2018a*; *Rajaraman et al., 2018b*). However, unlike natural images, medical images exhibit different visual characteristics including color, texture, shape, appearance, and their combinations, and exhibit low intra-class variance and high inter-class similarity (*Suzuki, 2017*), e.g., CXRs. Often, medical image collections are limited in size resulting in models overfitting and exhibiting poor generalization in real-world applications (*Srivastava et al., 2014*). Under these circumstances, the models can be retrained on a large-scale selection of modality-specific data, henceforth called *modality-specific transfer learning*. This strategy induces knowledge in the form of modality-specific feature representations toward improving generalization and performance for similar tasks. *Yadav, Passi & Jain (2018)* performed a coarse-to-fine knowledge transfer for CXR analysis by retraining a pretrained ResNet-50 (*He et al., 2016*) model on the National Institutes of Health (NIH) CXR dataset (*Wang et al., 2017*) and then repurposing the model to detect Tuberculosis (TB)-like manifestations in the NIH Shenzhen CXR dataset (*Jaeger et al., 2014*). Other literature on modality-specific transfer learning applied to CXR analysis is limited. In this regard, current research leaves room for progress in evaluating these strategies for improved learning and performance toward abnormality detection in CXRs.

CNNs are found to be sensitive to the noise present in the training data since they learn through stochastic optimization and error backpropagation. This leads to variance error and may result in overfitting since the learning algorithms may also model the random noise in the training set resulting in poor generalization to real-world data. A machine learning paradigm called ensemble learning reduces the prediction variance by combining constituent model predictions. This results in a model with superior performance compared to any individual constituent model (*Dietterich, 2000*). Several ensemble strategies including majority voting, simple averaging, weighted averaging, bagging, boosting, and stacking have been studied to minimize this variance and improve learning and generalization (*Rajaraman, Jaeger & Antani, 2019*). A literature survey reveals the use of several deep model ensembles for visual recognition tasks. *Krizhevsky, Sutskever & Hinton (2012)* used a CNN-based model called AlexNet and averaged the predictions of multiple model instances to achieve superior performance in the ImageNet Large Scale Visual Recognition Classification (ILSVRC) challenge. Since then, ensemble learning strategies have been used extensively. In CXR analysis, *Lakhani & Sundaram (2017)* used an averaging ensemble of CNNs to detect TB-like manifestations. The ensemble model for disease detection constructed with customized and ImageNet-pretrained CNNs achieved an area under the curve (AUC) of 0.99. We have previously proposed a stacked model ensemble with conventional handcrafted feature descriptors/classifiers and data-driven CNN models (*Rajaraman et al., 2018a*) toward detecting TB in chest radiographs. The individual models and their ensemble were evaluated on different CXR datasets. In the process, the ensemble
model achieved superior performance in terms of accuracy and AUC, compared to the other models and the state-of-the-art. *Islam et al. (2017)* trained different pretrained CNNs and constructed an averaging ensemble toward cardiomegaly detection using CXRs. The model ensemble detected the disease with 92% accuracy in comparison to hand-crafted feature descriptors/classifiers, achieving 76.5% accuracy. The ensemble of CNN models is observed to deliver superior performance in visual recognition tasks through reduced variance in their predictions and improved performance and generalization (*Rajaraman et al., 2019*).

CNN models are commonly perceived as black boxes with an increasing ask for an explanation of their decision-making process. A lack of explanation of learned interpretations has been viewed as a serious bottleneck in acceptance of the technology for medical screening or diagnostic applications (*Kim, Rajaraman & Antani, 2019*). Exploratory studies need to be performed to interpret and explain model behavior in clinical decision making. A survey of the literature reveals studies discussing the development of visualization strategies as a way to explain the learned representation in CNN models. *Zeiler & Fergus (2014)* proposed a deconvolution model to project the activations to the input pixel-space and visualize the input stimuli that excite the feature maps in the intermediate model layers. An ablation study was also performed to discover the performance contribution from different layers of the learned model to gain insight into its functioning. *Mahendran & Vedaldi (2015)* conducted a visual analysis of learned features through invert representations at different layers of a trained model. It was observed that the intermediate layers retained all information pertaining to different degrees of photometric and geometric image variance, thus providing an analysis of the visual information contained in image representations. *Zhou et al. (2016)* proposed a visualization strategy called class activation mapping (CAM) to localize image regions of interest (ROIs) that are relevant to an image category. However, the usage of this strategy is limited since it works only with CNN models having a fixed architecture. A generalized version, called gradient-weighted class activation mapping (Grad-CAM) was proposed in *Selvaraju et al. (2017)* to be applied to CNN models with varying architecture and hyperparameters. These tools rely on the prediction scores from a particular output node to classify the images to their respective categories. We have previously proposed a visualization strategy called class-selective relevance mapping (CRM) (*Kim, Rajaraman & Antani, 2019*) to localize discriminative ROIs in medical images. The algorithm measures the contributions of both positive and negative spatial elements in the feature maps in a trained model toward image classification. The visualization algorithm helped to generate modality-specific ROI mappings and demonstrated superior performance in comparison to the state-of-the-art in localizing the discriminative ROIs toward classifying the medical images belonging to different modalities.

Researchers have attempted to explain CNN model predictions and interpret the learned representations toward disease detection using CXRs. *Wang et al. (2017)* used a gradient-based region localization tool to spatially locate pneumonia in chest radiographs. An AUC of 0.633 was reported toward detecting eight different thoracic diseases. *Rajpurkar et al. (2017)* performed gradient-based ROI visualization and localization toward detecting

pneumonia in CXRs. They used a DenseNet-121 (*Huang et al., 2017*) model to estimate disease probability and obtained 0.768 AUC toward disease detection. An attention-guided CNN was proposed by *Guan et al. (2018)* to visualize and localize pneumonia infection in CXRs. The authors trained pretrained CNNs including ResNet-50 (*He et al., 2016*) and DenseNet-121 to learn the feature representations of disease-specific ROI. The authors reported an average AUC of 0.841 and 0.871 with ResNet-50 and DenseNet-121 model backbones, respectively. *Rajaraman et al. (2018a)* and *Rajaraman et al. (2018b)* used model-agnostic visualization tools and generated class-specific mappings to localize ROI that is considered relevant for detecting pneumonia and further categorizing bacterial and viral pneumonia using pediatric CXRs. The performance of the customized and pretrained CNN models was statistically validated. It was observed that the VGG-16 model achieved superior classification and localization performance with 96.2% and 93.6% accuracy, respectively, in distinguishing between bacterial and viral pneumonia.

This work aims to simplify the analysis in a binary triage classification problem that classifies CXRs into normal and abnormal categories. In this study, we propose modality-specific ensemble learning toward improving abnormality detection in CXRs. CNN models are trained on a large-scale CXR collection to learn modality-specific features. The learned knowledge is transferred and repurposed for detecting and localizing abnormalities using a different CXR collection. A custom, sequential CNN and a selection of pretrained models including VGG-16 (*Simonyan & Zisserman, 2015*), VGG-19 (*Simonyan & Zisserman, 2015*), Inception-V3 (*Szegedy et al., 2016*), Xception (*Chollet, 2017*), MobileNet (*Sandler et al., 2018*), DenseNet-121, and NASNet-mobile (*Pham et al., 2018*) are trained on the large-scale CheXpert CXR dataset (*Irvin et al., 2019*) to learn modality-specific features and repurposed for detecting and localizing abnormal CXR regions using the RSNA CXR dataset (*Shih et al., 2019*). The predictions of the CNNs are combined through different ensemble strategies including majority voting, simple averaging, weighted averaging, and stacking, with an aim to reduce prediction variance and sensitivity to the training data, the learning algorithm, and to improve overall performance and generalization. Also, we employ visualization techniques to interpret the significant features that helped in image categorization. The learned behavior of individual models and their ensemble is visualized using the CRM visualization tool (*Kim, Rajaraman & Antani, 2019*). To the best of our knowledge, this is the first study applied to CXR analysis that proposes a combination of modality-specific knowledge transfer and ensemble learning and evaluates ensemble-based disease ROI localization. We evaluate the localization performance of the model ensembles using Intersection of Union (IoU) and mean average precision (mAP) metrics.

## MATERIAL AND METHODS

### Data collection and preprocessing

The publicly available CheXpert (*Irvin et al., 2019*) and RSNA CXR (*Shih et al., 2019*) collections are used in this retrospective study. The characteristics of the datasets are mentioned herewith:

CheXpert CXR collection: *Irvin et al. (2019)* have collected 223,648 CXRs from 65,240 patients at Stanford Hospital, California, USA. The CXRs are captured in frontal and lateral

**Table 1 Datasets characteristics.** The datasets are split at the patient-level into 80% for training and 20% for testing; 10% of the training data is randomly allocated for validation.

| Dataset | Abnormal | | | Normal | | | Type |
|---|---|---|---|---|---|---|---|
| | Train | Test | Total | Train | Test | Total | |
| CheXpert (Frontal) | 139,383 | 34,846 | 174,229 | 13,600 | 3,400 | 17,000 | JPG |
| RSNA CXR | 14,266 | 3,567 | 17,833 | 7,080 | 1,771 | 8,851 | DICOM |

projections and labeled for normal and 14 different thoracic disease manifestations based on clinical relevance and conforming to the Fleischner Society's glossary. The authors developed an automatic rule-based labeling tool to extract clinical observations from the radiological reports and used them toward image labeling. In this study, we grouped the CXRs with disease manifestations to construct the abnormal class.

Radiological Society of North America (RSNA) CXR collection: The dataset has been released as a part of the RSNA Kaggle pneumonia detection challenge, jointly organized by the radiologists from RSNA, Society of Thoracic Radiology (STR), and the NIH (*Shih et al., 2019*). The dataset includes 26,684 normal and abnormal frontal CXRs and is made available in DICOM format at $1,024 \times 1,024$ spatial resolution. Each CXR image carries one of the three labels: normal, not-abnormal/not-opacity, and lung opacity. We grouped the not-abnormal/not-opacity and lung opacity labeled images to construct the abnormal class. Table 1 shows the distribution of data across the different categories for these datasets. Of the 3,567 abnormal test images for the RSNA CXR dataset, the authors (*Shih et al., 2019*) have made available the ground-truth (GT) bounding boxes for only 1,241 abnormal images that contain pneumonia-related opacities.

The datasets have been split at the patient-level into 80% for training and 20% for testing for the different stages of learning performed in this study. We have randomly allocated 10% of the training data for validation.

## Lung segmentation and bounding box cropping

The CXRs contain regions that do not contribute to diagnosing lung abnormalities. Hence, CNN models may learn irrelevant features that impact decision making. Under these circumstances, semantic segmentation is performed using algorithms like U-Net (*Ronneberger, Fischer & Brox, 2015*) to perform pixel-level labeling where each image pixel is labeled to produce segmentation maps. The U-Net consists of a contraction/encoder with convolutional layers to capture image context and a symmetrical expansion/decoder with transposed convolutions to perform localization. The fully-connected network architecture accepts images of any size since they do not have dense layers. A dropout U-Net is used in this study (*Novikov et al., 2018*) where a dropout layer is placed after the convolutional layers in the encoder. The addition of dropout layers aid in decreasing generalization error and enhance learning by providing restrictive regularization (*Srivastava et al., 2014*). We used Gaussian dropout since it is observed to perform superior to the classical approach that uses Bernoulli distribution to drop neural units (*Srivastava et al., 2014*). After empirical evaluations, a dropout ratio of 0.2 is used in this study. Figure 1 shows the dropout U-Net architecture and the segmentation pipeline is shown in Fig. 2. The U-Net model is trained

on CXR images and their associated lung masks made publicly available by *Candemir et al. (2015)*. The images and lung masks are augmented on-the-fly during model training with an aim to reduce overfitting and improve generalization. We used sigmoid activation to ensure the mask pixels lie in the range [0 1]. Early stopping and callbacks are used to check the training performance at every epoch and store the best model weights for mask generation. The model outputs the learned lung masks at $256 \times 256$ spatial resolution for the CheXpert and RSNA CXR datasets used in this study. The generated lung masks are used to delineate the lung boundaries that are cropped to the size of a bounding box, accommodating the lung pixels. The lung crops are rescaled to $256 \times 256$ pixel dimensions, the lung bounding box coordinates are recorded and stored for further evaluations.

The cropped lung boundaries are preprocessed as follows: The images are (a) passed through a median filter with a $3 \times 3$ kernel to preserve edges and remove noise; (b) rescaled to the pixel range [0 1]; and (c) mean-normalized and standardized for identical distribution of the extracted features.

## Models and computational resources

We evaluated the performance of the following CNN models at different stages of learning performed in this study: (a) Custom CNN; (b) VGG-16; (c) VGG-19; (d) Inception-V3; (e) Xception; (f) DenseNet-121; (g) MobileNet; and (h) NASNet-mobile. The custom CNN is constructed as a linear stack of depth-wise separable convolution, nonlinear activation, pooling, and dense layers. Depth-wise separable convolution applies the convolution operation to individual channels, followed by a point-wise convolution with $1 \times 1$ kernels. These operations are shown to result in fewer model parameters and reduced overfitting as compared to conventional convolutions (*Chollet, 2017*). The architectural framework of the custom CNN is shown in Fig. 3.

The convolutional block consists of a separable convolution layer, followed by batch normalization and ReLU non-linearity layers. We added padding to the separable convolutional layers to ensure the feature map dimensions of the intermediate layers match the original input size. We used $5 \times 5$ kernels for all separable convolutional layers. Each convolutional block is followed by a max-pooling layer; the number of kernels is increased by a factor of two in the succeeding convolutional blocks to ensure the computation roughly remains the same across the separable convolutional layers. The model is added with a global average pooling (GAP) layer, followed by dropout (ratio = 0.5), and a final dense layer with Softmax activation to output prediction probabilities.

We performed Bayesian learning (*Mockus, 1974*) to optimize the architecture and hyperparameters of the custom CNN toward the current task. The optimization procedure involves minimizing a Gaussian process model of an objective function. Bayesian optimization is performed to find the optimal values for the parameters including learning rate, depth, L2-weight decay, and momentum. The "depth" parameter controls model depth. The model has three sections, each with "depth" identical convolutional blocks. The total number of convolutional blocks is "$3 \times$ depth". We initialized the number of separable convolutional kernels to roundup(image size/$\sqrt{\text{depth}}$). The search ranges for the learning rate, depth, L2-weight decay, and momentum are set to [1e−5.1e−2], [1 4], [1e−8.1e−3],

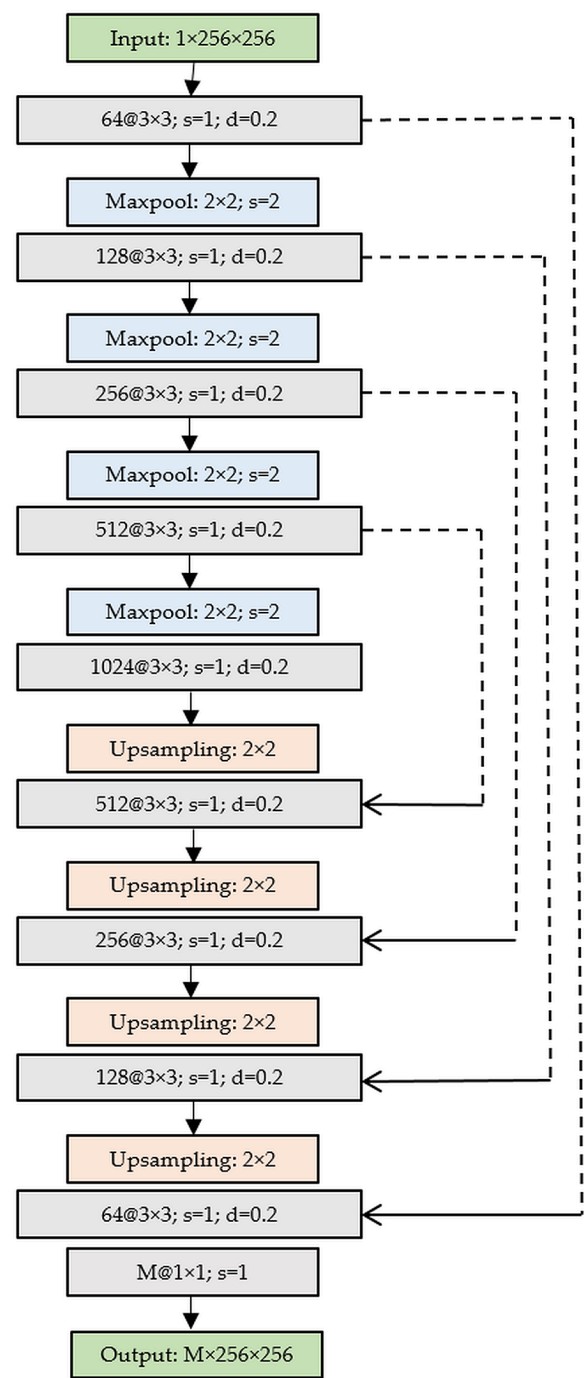

**Figure 1** **Architecture of the dropout U-Net.**

and [0.85 0.99] respectively. The objective function trains the model within the search ranges specified for the optimizable parameters and the Bayesian-optimized model parameters for the least validation error are recorded. Based on empirical observations, we performed 30 objective function evaluations toward hyperparameter optimization. The final model

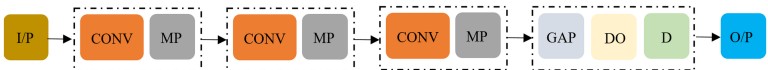

**Figure 2** Segmentation pipeline showing mask generation using a dropout U-NET and lung boundary cropping.

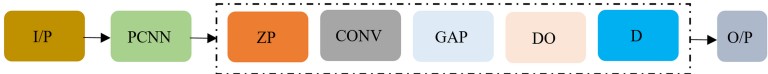

**Figure 3** Architecture of the custom CNN model (I/P, Input; CONV, Convolutional block; MP, Max-pooling; GAP, Global average pooling; DO, Dropout; D, Dense layer with Softmax activation).

**Figure 4** Architecture of the pretrained CNNs (I/P, Input; PCNN, Truncated pretrained CNNs; ZP, Zero-padding; CONV, Convolution; GAP, Global Average Pooling; DO, Dropout; D, Dense layer with Softmax activation).

with the optimized parameters is trained, validated, and tested through stochastic gradient descent (SGD) optimization at different stages of learning discussed in this study to estimate the generalization error and classification performance.

The pretrained CNN models are instantiated with the ImageNet weights and truncated at their fully-connected layers. The truncated models are appended with the following layers: (a) zero-padding; (b) convolutional layer with 3 × 3 kernels and 1024 feature maps; (c) GAP; (d) dropout (dropout ratio = 0.5); and (e) dense layer with Softmax activation. The customized architecture of the pretrained CNNs used in this study is as shown in Fig. 4.

The hyperparameters of the pretrained CNNs including momentum, learning rate, and L2-weight decay, in modality-specific transfer learning and coarse-to-fine-learning, are optimized through a randomized grid search (*Bergstra & Bengio, 2012*). The search ranges are set to [0.85 0.99], [1e−9 1e−2], and [1e−10 1e−3] for the momentum, learning rate, and L2-decay respectively. We retrained the models with smaller weight updates through SGD optimization to minimize the categorical cross-entropic loss toward CXR classification. The magnitude of the weight updates is kept small to improve generalization. We used class weights to assign higher weights to the underrepresented class with an aim to prevent model bias and overfitting (*Johnson & Khoshgoftaar, 2019*). The learning rate is reduced whenever the validation performance plateaued. Callbacks are used to check the models' internal states, checkpoints are stored for every epoch, early stopping is performed to prevent overfitting, and the best model weights are stored to perform hold-out testing.

The models in modality-specific transfer learning, coarse-to-fine-learning, and ensemble learning are evaluated in terms of the following performance metrics: (a) accuracy; (b) AUC; (c) sensitivity; (d) specificity; (e) F measure; and (f) Matthews correlation coefficient (MCC). We used Keras API with Tensorflow backend and CUDA/CUDNN libraries for GPU acceleration. Matlab R2018b® is used for custom CNN optimization. The models are trained and evaluated on an Ubuntu Linux system with 64GB RAM and NVIDIA 1080Ti GPU.

## Modality-specific transfer learning

We propose modality-specific transfer learning where the pretrained CNNs with ImageNet weights and the custom CNN with random weight initializations are trained end to end and evaluated on the large-scale CheXpert data set to classify abnormal and normal CXRs, thereby making all the weight layers specific to the CXR modality. The idea behind this approach is to induce knowledge pertaining to a large-scale selection of CXR lung abnormalities and provide possible hints to how the abnormal and normal lungs look like. The CheXpert dataset includes normal CXRs and abnormal images containing the following abnormalities: (a) enlarged cardio-mediastinum; (b) cardiomegaly; (c) lung opacity; (d) lung lesion; (e) edema; (f) consolidation; (g) pneumonia; (h) atelectasis; (i) pneumothorax; (j) pleural effusion; (k) pleural other; and (l) fracture. CXRs with disease manifestations are grouped to form the abnormal class and used to solve the binary classification task. The learned knowledge is then transferred and repurposed to perform CXR abnormality detection using the RSNA CXR dataset, with an aim to improve models' adaptation, performance, and generalization. During model training, we augmented the CXR images with horizontal, vertical translations and rotations with an aim to improve generalization and reduce overfitting to the training data. The modality-specific trained CNNs are henceforth called as *coarse models*.

## Coarse-to-fine learning

For coarse-to-fine learning, we instantiated the convolutional part of the coarse models and extracted the features from different intermediate layers with an aim to identify the optimal layer for feature extraction toward the current task. Pretrained models differ in their depth and learn different feature representations pertaining to their depth. Deeper models are not always optimal for all the tasks, especially in the medical domain, where the data distribution is different compared to stock photographic images. In this regard, it is indispensable to identify the layer in the individual models to extract features that deliver the best performance toward classifying the classes under study. We used class weights to penalize the majority class with an aim to prevent model bias. The retrained models are henceforth called as *fine* models and the process, *coarse-to-fine learning*. The workflow is shown in Fig. 5. As in modality-specific transfer learning, the data is augmented with horizontal, vertical translations and rotations during model training. The naming conventions for the models' layers follow that available from the Keras library. The models

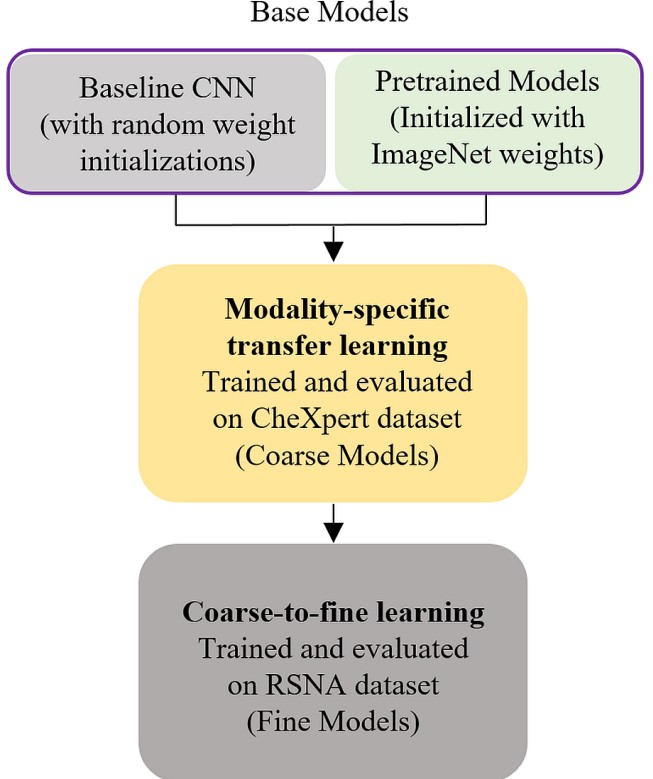

Base Models

**Figure 5** Modality-specific knowledge transfer workflow.

are trained through SGD optimization and the best model weights are stored to perform hold-out testing with the RSNA CXR test set.

## Ensemble learning

We constructed ensembles of the top-7 fine models with an aim to reduce (i) prediction variance, (ii) sensitivity to the specifics of the training data, (iii) model overfitting, and (iv) improve performance and generalization. We performed majority voting, simple averaging, weighted averaging, and stacking with an aim to build a predictive model with reduced prediction variance and improved performance as compared to any individual constituent model (*Dietterich, 2000*). In majority voting, the predictions from the individual models are considered vote; the prediction with the maximum votes is considered the final prediction. In simple averaging, we computed the average of the constituent model predictions to arrive at the final prediction. The weighted averaging ensemble is an extension of simple averaging where the constituent model predictions are assigned different weights based on their classification performance. Model stacking is an ensemble method that performs second-level learning using a meta-learner that learns to combine the predictions of the individual models, otherwise called the base-learners (*Dietterich, 2000*). A stacked model is considered a single model where the predictions of the individual base-learners are fed

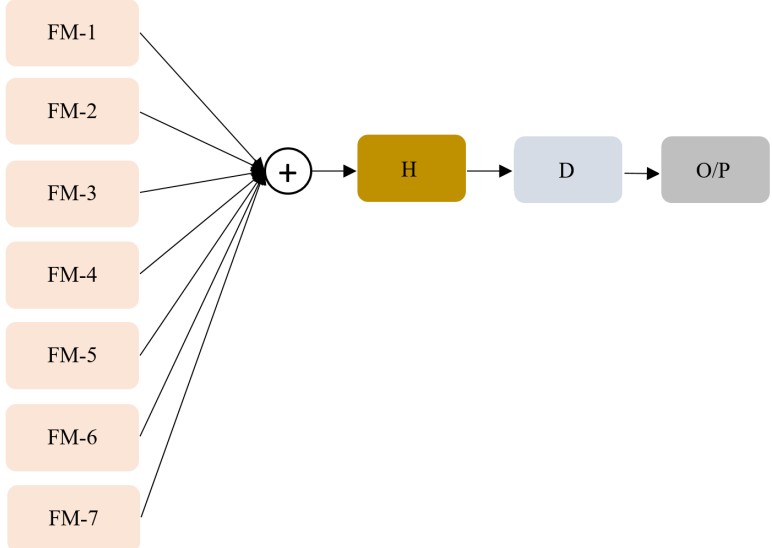

**Figure 6 Stacked generalization.** FM (1-7) denotes fine models (base learners) with their outputs concatenated to construct a single 14-element vector from the two class-probabilities (abnormal and normal) predicted by each of the fine models; A single hidden layer is represented by 'H' with 14 neurons that interprets the input, and the Dense layer is represented by 'D' with Softmax activation that outputs probabilistic predictions.

to the meta-learner and are embedded into the multi-headed network architecture. The stacking workflow is shown in Fig. 6.

Model stacking performs learning at distinct stages: (a) Stage-0: The base-learners are trained, validated, and tested using the RSNA CXR test set to output predictions; (b) Stage-1: A neural-network-based meta-learner learns to combine the predictions of individual base-learners. The meta-learner consists of a hidden layer with 14 neurons that interprets the input from the base-learners and a dense layer with Softmax activation to output the prediction probabilities. We freeze the trainable weights in the individual base-learners and train the meta-learner with an aim to output improved predictions as compared to any individual base-learner.

## Statistical analysis

We performed statistical testing to investigate the existence of a statistically significant difference in performance between the models in different stages of learning discussed in this study. Confidence intervals (CI) are used in applied DL to present the skill of predictive models by measuring the precision of an estimate through the margin of error. A relatively precise estimate is inferred by a short CI and therefore a smaller error margin. A larger error margin is inferred by a long CI and therefore low precision (*Gulshan et al., 2016*). In this regard, we computed the 95% CI for the AUC values obtained by the models in modality-specific transfer learning, coarse-to-fine learning, and ensemble learning, to be the Wilson score interval which corresponds to separate 2-sided CI with individual
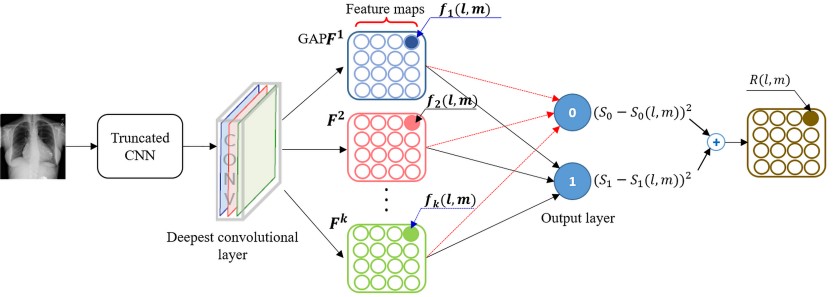

**Figure 7  A schematic representation showing the calculation of class-selective relevance mapping (CRM) from a CNN-based DL model.**

coverage probabilities of sqrt(0.95). Statistical significance and simultaneous 2-sided CI are computed using the StatsModels version 0.11.0 and SciPy version 1.4.1 python packages.

# MODEL VISUALIZATION

## Class-selective relevance mapping

The learned behavior of the individual models and their ensembles is visualized based on the CRM visualization tool (*Kim, Rajaraman & Antani, 2019*) to localize discriminative ROI showing abnormal CXR regions and explain models predictions. CRM visualization algorithm measures the significance of the activations in the feature maps at the deepest convolution layer of a CNN model to highlight the most discriminative ROI in the input image at the spatial location $(x, y)$. A prediction score $S_c$ is calculated at each node $c$ in the output layer. Another prediction score $S_c(l, m)$ iscalculated after removing a spatial element $(l, m)$ in the feature maps from the deepest convolutional layer. The CRM $R(l, m) \in \mathbb{R}^{u \times v}$ is defined as a linear sum of incremental MSE between $S_c$ and $S_c(l, m)$ calculated from all nodes in the output layer of the CNN model.

$$R(l, m) = \sum_{c=1}^{N} \{(S_c - S_c(l, m))\}^2 \tag{1}$$

Accordingly, a spatial element having a large CRM score can be considered as playing an important role in the classification process since removing that node results in a significant increase in the mean squared error (MSE) at the output layer. Figure 7 illustrates a conceptual workflow for measuring the CRM score from a CNN model and is simplified for the purposes of reader understanding to the case of a two-class problem. Since DL is a discriminative learning process, an important element in the feature maps from the deepest convolution layer would make not only a positive contribution to increasing the prediction score at the output node representing the desired class but also a negative contribution to decreasing the prediction score at the remaining output nodes. This maximizes the gap between these prediction scores. Since the CRM is based on the incremental MSE calculated from all output nodes, the resulting ROI is determined to be class-discriminative.
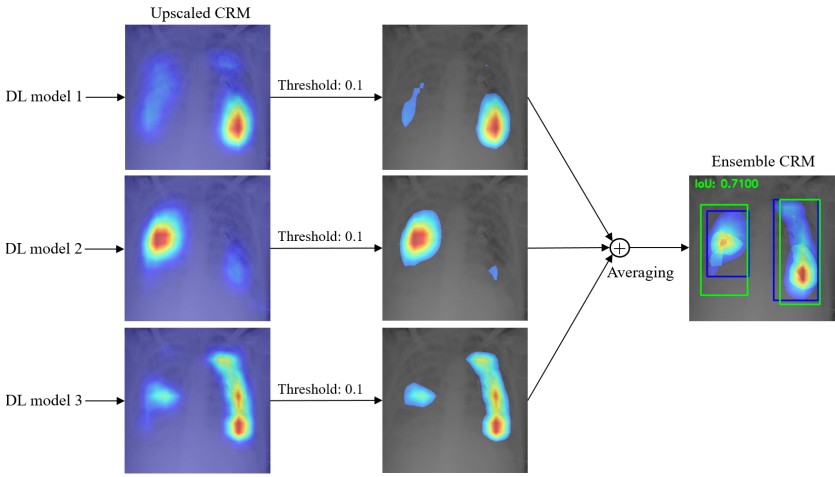

Upscaled CRM

DL model 1 → Threshold: 0.1

DL model 2 → Threshold: 0.1

DL model 3 → Threshold: 0.1

Averaging

Ensemble CRM

**Figure 8** **Workflow showing an ensemble CRM constructed from three individual CRMs.**

## Ensemble CRM

An ensemble CRM is created by combining and averaging multiple CRMs generated from different CNN models; the workflow for a selection of three CNN models is shown in Fig. 8. Since each CNN model generates a different CRM size depending on the spatial dimensions of the feature maps from its deepest convolution layer, we normalized the size of all individual CRMs by upscaling them to the size of the input image. We removed the mapping score value below 10% of the maximum score in each CRM to minimize a possible noisy influence of a very low mapping score during the ensemble process. The thresholded CRMs are combined through simple averaging to construct the ensemble CRM by displaying mapping score values above 10% of its maximum score. This ensemble CRM is compensated for the error of missing ROI from an individual CNN model with an aim to improve the overall localization performance.

To demonstrate the effectiveness of our ensemble strategy, we created three ensemble CRMs by combining the top-3, top-5, and top-7 performing CNN models respectively and quantitatively compared their visual localization performance with each other and with individual CRMs in terms of IoU and mAP metrics. IoU is an evaluation metric to measure the accuracy of object detection and is defined as "*area of the overlap*"/"*area of the union*" between the GT bounding box and the predicted bounding box for a given input image (*Everingham et al., 2015*). The mAP score is calculated by taking the mean AP over the IoU thresholds as detailed in (*Lin et al., 2014*).

## RESULTS

### Performance metrics evaluation

Table 2 shows the optimal values obtained for the hyperparameters through (i) Bayesian optimization for the custom CNN and (ii) randomized grid search for the pretrained CNNs. Table 3 shows the performance achieved by the coarse models using the CheXpert test set. It is observed that the VGG-16 model demonstrated superior performance in all

**Table 2  Optimized hyperparameter values for the CNN models.** Bayesian learning was performed to optimize the architecture and hyperparameters of the custom and pretrained CNNs by minimizing a Gaussian process model of an objective function that trains the models within the search ranges specified for the optimizable parameters including learning rate, momentum, network depth, and L2-decay and the Bayesian-optimized model parameters for the least validation error are recorded.

| Model | Optimal values | | | |
|---|---|---|---|---|
| | Learning rate | Momentum | Depth | L2-decay |
| Custom CNN | 1e−3 | 0.90 | 1 | 1e−6 |
| Pretrained CNNs | 1e−4 | 0.95 | – | 1e−6 |

**Table 3  Performance metrics achieved by the coarse models using the CheXpert test set.** The coarse models are initialized with ImageNet pretrained weights (conventional transfer learning) and trained end-to-end to learn CXR modality-specific weights using the CheXpert data set to classify the CXRs into normal and abnormal classes. The custom CNN is initialized with random weights. Data in parenthesis are 95% CI for the AUC values that were calculated to be the Wilson score interval which corresponds to separate 2-sided confidence intervals with individual coverage probabilities of sqrt(0.95).

| Model | Accuracy | AUC | Sensitivity | Specificity | F measure | MCC |
|---|---|---|---|---|---|---|
| Custom CNN | 0.8018 | 0.8898 (0.8813, 0.8983) | 0.9030 | 0.5980 | 0.7952 | 0.5356 |
| VGG-16 | **0.8904** | **0.9649 (0.9599, 0.9699)** | **0.9173** | **0.8448** | **0.8904** | **0.7530** |
| VGG-19 | 0.8799 | 0.9432 (0.9369, 0.9495) | 0.9115 | 0.8165 | 0.8798 | 0.7288 |
| Inception-V3 | 0.8835 | 0.9571 (0.9516, 0.9626) | 0.9028 | 0.8363 | 0.8840 | 0.7402 |
| Xception | 0.8720 | 0.9401 (0.9337, 0.9465) | 0.9005 | 0.8148 | 0.8723 | 0.7126 |
| DenseNet-121 | 0.8839 | 0.9493 (0.9434, 0.9552) | 0.9140 | 0.8233 | 0.8838 | 0.7378 |
| MobileNet | 0.8797 | 0.9456 (0.9395, 0.9517) | 0.9073 | 0.8244 | 0.8799 | 0.7295 |
| NASNet-mobile | 0.8824 | 0.9552 (0.9496, 0.9608) | 0.9045 | 0.8380 | 0.8828 | 0.7369 |

**Notes.**
Bold values indicate superior performance.

performance metrics. The VGG-16 model demonstrated superior AUC values with a short CI, signifying a smaller error margin and therefore offering a precise estimate as compared to the other CNN models. The architectural depth of the VGG-16 model remained optimal to learn the hierarchical feature representations to improve performance with the CheXpert test set, as compared to the other CNN models.

Table 4 lists the empirically determined, best-performing model layers while performing coarse-to-fine learning. The performance metrics achieved through extracting the features from these intermediate layers that helped achieve superior performance toward classifying normal and abnormal CXRs using the RSNA CXR test set are shown in Table 5. The baseline models refer to the custom CNN and ImageNet-pretrained CNN models that are trained end to end and evaluated on the RSNA CXR data set. This is the conventional transfer learning reported in the literature. It is observed from Table 5 that the features extracted and classified from the different intermediate layers of the coarse models to create fine models delivered better performance than their baseline counterparts. This performance improvement could be attributed to the fact that modality-specific transfer learning from a large-scale CXR collection helped induce knowledge pertaining to a large selection of abnormal lung manifestations and normal lungs and improved classification performance with a related classification task using the RSNA CXR dataset.

**Table 4** **Candidate CNN layers from the coarse models showing superior performance with the RSNA CXR test set.** The performance metrics achieved through feature extraction from these intermediate layers of the coarse models helped to achieve superior performance toward classifying normal and abnormal CXRs using the RSNA CXR test set. The naming conventions for the models layers follow that available from the Keras DL library.

| Model | Feature extraction layer |
| --- | --- |
| Custom CNN | Conv3 |
| VGG-16 | Block5-conv3 |
| VGG-19 | Block4-pool |
| Inception-V3 | Mixed3 |
| Xception | Add-7 |
| DenseNet-121 | Pool3-conv |
| MobileNet | Conv-pw-6-relu |
| NASNet-mobile | Activation-129 |

**Table 5** **Performance metrics computed with baseline CNNs (conventional transfer learning) and feature extraction from the optimal intermediate layers of the coarse models to create fine models using the RSNA CXR test set.** Baseline models refer to the custom and ImageNet-pretrained CNN models that are trained end to end and evaluated on the RSNA CXR data set (conventional transfer learning). To create fine models, the CXR modality-specific coarse models are instantiated with their modality-specific weights and features are extracted from the optimal intermediate layers. These models are evaluated using the RSNA CXR dataset to classify them into normal and abnormal classes. Data in parenthesis are 95% CI for the AUC values that were calculated to be the Wilson score interval which corresponds to separate 2-sided confidence intervals with individual coverage probabilities of sqrt(0.95).

| Model | | Accuracy | AUC | Sensitivity | Specificity | F measure | MCC |
| --- | --- | --- | --- | --- | --- | --- | --- |
| Custom CNN | Baseline | 0.8012 | 0.8852 (0.8766, 0.8938) | 0.8901 | 0.5978 | 0.7947 | 0.5347 |
| | Fine | 0.8442 | 0.9171 (0.9097, 0.9245) | 0.9023 | 0.7718 | 0.8441 | 0.6478 |
| VGG-16 | Baseline | 0.8610 | 0.9153 (0.9078, 0.9228) | 0.8907 | 0.8057 | 0.8456 | 0.7214 |
| | Fine | **0.8946** | **0.9649 (0.9599, 0.9699)** | **0.9225** | **0.8912** | **0.9201** | **0.7564** |
| VGG-19 | Baseline | 0.8602 | 0.9057 (0.8978, 0.9136) | 0.8726 | 0.7937 | 0.8314 | 0.7178 |
| | Fine | 0.8921 | 0.9647 (0.9597, 0.9697) | 0.9216 | 0.8351 | 0.9182 | 0.7541 |
| Inception-V3 | Baseline | 0.8503 | 0.8962 (0.8880, 0.9044) | 0.8658 | 0.8002 | 0.8327 | 0.7181 |
| | Fine | 0.8821 | 0.9639 (0.9588, 0.9690) | 0.8951 | 0.8613 | 0.9105 | 0.7417 |
| Xception | Baseline | 0.8672 | 0.9372 (0.9306, 0.9438) | 0.8905 | 0.8137 | 0.8716 | 0.7118 |
| | Fine | 0.8791 | 0.9546 (0.9490, 0.9602) | 0.9212 | 0.8238 | 0.9087 | 0.7203 |
| DenseNet-121 | Baseline | 0.8532 | 0.9078 (0.9000, 0.9156) | 0.8582 | 0.8326 | 0.8458 | 0.7132 |
| | Fine | 0.8873 | 0.9548 (0.9492, 0.9604) | 0.9015 | 0.8497 | 0.9125 | 0.7421 |
| MobileNet | Baseline | 0.8558 | 0.9082 (0.9004, 0.9160) | 0.8621 | 0.8316 | 0.8447 | 0.7121 |
| | Fine | 0.8801 | 0.9576 (0.9521, 0.9631) | 0.8923 | 0.8582 | 0.9079 | 0.7358 |
| NASNet-mobile | Baseline | 0.8501 | 0.9181 (0.9107, 0.9255) | 0.8325 | 0.8615 | 0.8462 | 0.7119 |
| | Fine | 0.8740 | 0.9544 (0.9488, 0.9600) | 0.8642 | 0.8270 | 0.9012 | 0.7311 |

**Notes.**
Bold values indicate superior performance.

We performed ensembles of the predictions of the top-7 fine CNN models through majority voting, simple averaging, weighted averaging, and stacking to classify the CXRs in the RSNA CXR test set into normal and abnormal categories. The performance metrics achieved with the different model ensembles are shown in Table 6. In weighted averaging, we awarded high/low importance to the predictions by assigning higher weights to more

**Table 6** **Performance metrics achieved with different model ensemble strategies using the RSNA CXR test set.** Data in parenthesis are 95% CI for the AUC values that were calculated to be the Wilson score interval which corresponds to separate 2-sided confidence intervals with individual coverage probabilities of sqrt(0.95).

| Method | Accuracy | AUC | Sensitivity | Specificity | F measure | MCC |
|---|---|---|---|---|---|---|
| Majority voting | 0.8923 | – | 0.9127 | 0.8526 | 0.9178 | 0.7654 |
| Averaging | 0.8964 | 0.9551 (0.9495, 0.9607) | 0.9118 | 0.8599 | 0.9201 | 0.7712 |
| Weighted-averaging | **0.9163** | **0.9747 (0.9704, 0.9790)** | **0.9249** | **0.8842** | **0.9286** | **0.7895** |
| Stacking | 0.8907 | 0.9552 (0.9496, 0.9608) | 0.8944 | 0.8527 | 0.9134 | 0.7601 |

**Notes.**
Bold values indicate superior performance.

**Table 7** **IoU and mAP for CRMs from each individual CNN model obtained by averaging IoU and mAP of all abnormal images having GT bounding box information in the RSNA CXR test set.**

| Metrics | VGG-16 | VGG-19 | Xception | Inception-V3 | MobileNet | NASNet-mobile | DenseNet-121 |
|---|---|---|---|---|---|---|---|
| IoU | 0.383 | 0.357 | 0.377 | 0.351 | 0.368 | 0.375 | 0.355 |
| mAP@[0.1 0.6] | 0.377 | 0.341 | 0.388 | 0.348 | 0.352 | 0.382 | 0.317 |

accurate base-learners. We empirically found that VGG-16 and VGG-19 models delivered superior performance compared to other models. Thus, we assigned weights of [0.25, 0.25, 0.1, 0.1, 0.1, 0.1, 0.1] to the predictions of VGG-16, VGG-19, Inception-V3, Xception, DenseNet-121, MobileNet, and NASNet-mobile models respectively. We observed that weighted averaging outperformed majority voting, simple averaging, and stacking in all performance metrics. As well, the tests for statistical significance demonstrated that the weighted averaging ensemble demonstrated superior values for AUC with a shorter CI and hence a smaller error margin than the other model ensembles. Also, considering the balance between precision and recall, as demonstrated by the F-measure and MCC, the weighted averaging ensemble outperformed the other methods.

## Visual localization evaluation

We evaluated the localization performance of CRMs generated from each of the top-7 fine CNN models in detecting abnormalities using the RSNA CXR test set. Table 7 shows the IoU and mAP scores resulting from averaging individual IoUs and mAPs calculated from a total of 1,241 abnormal images in the RSNA CXR test set that have GT bounding box information. Here, mAP is calculated by taking the mean AP over 10 IoU threshold values within the range [0.1 0.6] (denoted as mAP@[0.1 0.6]).

It is observed that VGG-16, Xception, and NASNet-mobile models are the top-3 performing in ROI detection and localization. Figure 9 shows the corresponding precision–recall curves with respect to different IoU threshold values, where each curve is generated by varying the confidence score threshold. The confidence score for ROI detection is defined as the highest heatmap score within the predicted bounding box weighted by the classification score at the output node in a given CNN model. Accordingly, each curve represents a plot of precision and recall when an ROI detection having IoU and confidence scores higher than their corresponding threshold values are only considered true positive predictions. The AP score of each curve is then calculated by taking the average value of

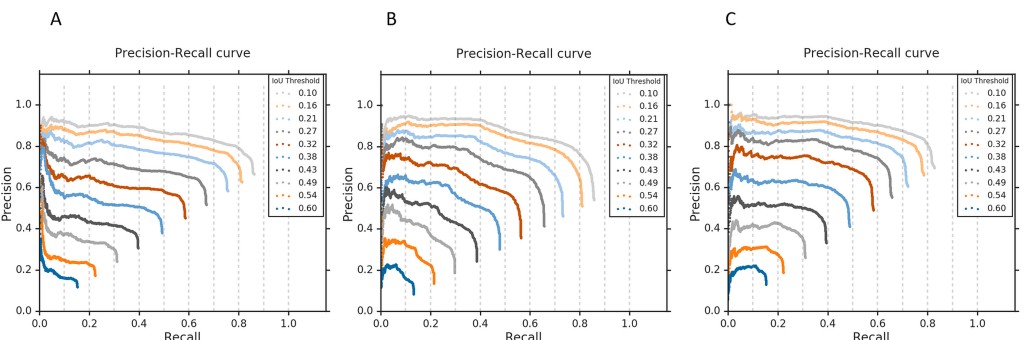

**Figure 9 Precision-recall curves with respect to the different IoU thresholds for the top-3 performing CNNs.** (A) VGG-16, (B) Xception, and (C) NASNet-mobile, calculated from all abnormal images having GT bounding box information in the RSNA CXR test set.

**Table 8 IoU and mAP for ensemble DL models reported using the RSNA CXR test set.**

| Metrics | Ensemble-3 | Ensemble-5 | Ensemble-7 |
|---|---|---|---|
| IoU | 0.430 | **0.433** | 0.432 |
| mAP@[0.1 0.6] | 0.420 | **0.447** | 0.434 |

**Notes.**
Bold values indicate superior performance.

the precision across all recall values. The above-mentioned IoU threshold value range of [0.1 0.6] was determined based on these precision–recall curves to avoid very poor and high precision and recall rates, thereby calculating the mAP score correctly reflecting the localization performance of each DL model.

We then calculated the IoU and mAP scores for three ensemble CRMs: (a) Ensemble-3; (b) Ensemble-5; and (c) Ensemble-7. These ensemble CRMs are generated by averaging the CRMs from the top-3, top-5, and top-7 performing CNN models that are selected based on the IoU and mAP scores as shown in Table 6. The models involved in different ensemble CRMs are: (a) Ensemble-3 (VGG-16, Xception, NASNet-mobile); (b) Ensemble-5 (VGG-16, Xception, NASNet-mobile, Inception-V3, MobileNet); and (c) Ensemble-7 (VGG-16, Xception, NASNet-mobile, Inception-V3, MobileNet, VGG-19 and DenseNet-121). As observed from Table 8, the ensemble CRMs yield significantly higher IoU and mAP scores as compared to individual CRMs. Among the ensemble CRMs, Ensemble-5 demonstrated superior performance for IoU and mAP metrics. This indicated that combining more than the top-5 CNN models does not improve the overall localization performance further and rather saturates for this study. Figure 10 shows the precision–recall curves of the ensemble CRMs from which their mAP scores are calculated. Figure 11 shows an example of Ensemble-5 CRM demonstrating the effectiveness of our ensemble approach in localizing abnormal ROIs within a given CXR image from the RSNA CXR test set.

We observed that each individual CRM highlights different areas as their ROIs for the given image and the bounding boxes surrounding these ROIs have low IoU scores even though all ROIs detected are actually located within the GT bounding boxes. On the other

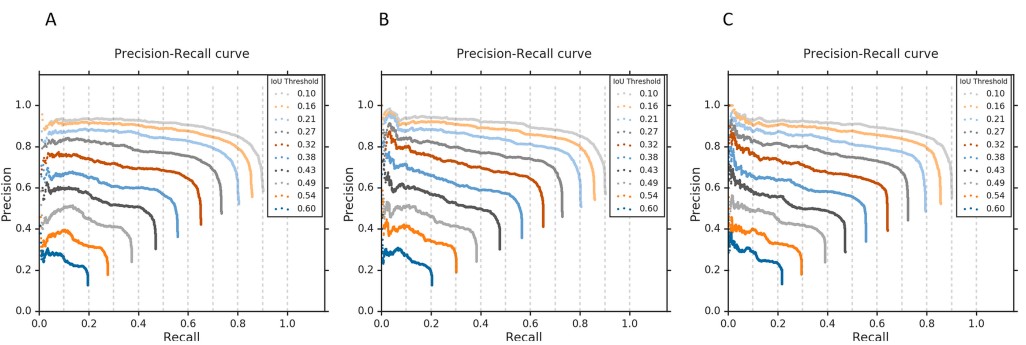

**Figure 10** Precision-recall curves with respect to the different IoU thresholds for (A) Ensemble-3, (B) Ensemble-5, and (C) Ensemble-7 models, calculated from all abnormal images having GT bounding box information in the RSNA CXR test set.

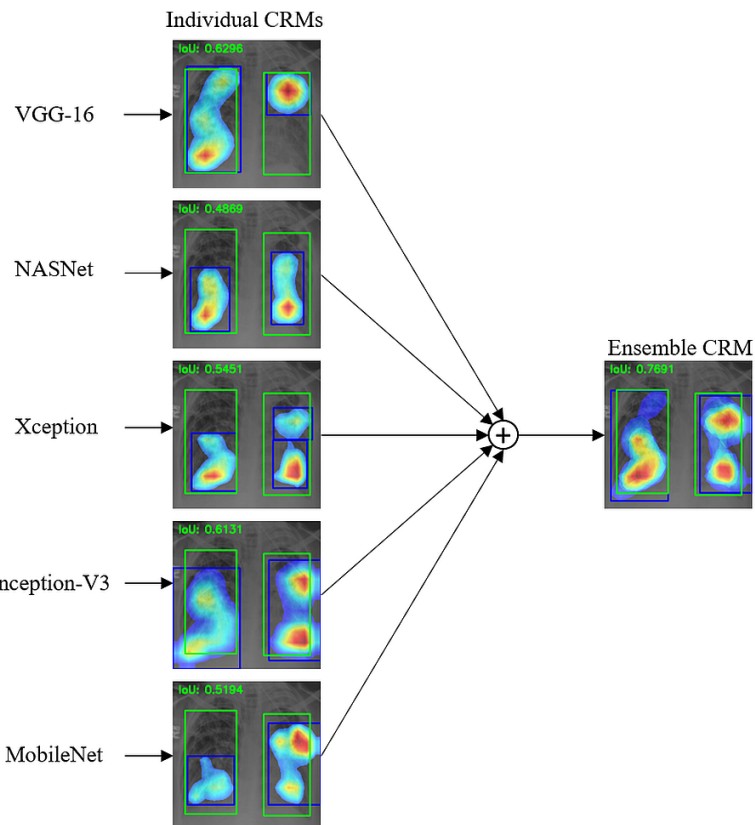

**Figure 11** An example of ensemble CRM combining the top-5 CNN models.

hand, the proposed ensemble CRM highlights ROIs resulting from averaging the ROIs from individual CRMs. The ensemble CRM visually and quantitatively demonstrates superior performance in objection detection and localization than any individual constituent model; the resulting ROIs are found to have their bounding boxes more closely overlapped with

the GT boxes and significantly improved IoU scores, compared to those from individual CRMs. Therefore, we demonstrate that our ensemble approach improves not only the classification performance but also the object detection performance overall.

## DISCUSSION

Since the coarse models have already learned the ability to classify the normal and abnormal CXRs with a large-scale dataset with a range of data distributions, the learned weights served as a promising initialization for a related abnormality classification task in a dataset with a different distribution, as compared to ImageNet weights learned from stock photographic images that are visually distinct from medical images. The size of the CheXpert data set is much larger than the RSNA CXR data set, thus transferring knowledge to the CXR-modality on the first hand allowed for improved initialization and adaptation of the CNNs that are subsequently trained and evaluated for a related classification task. The VGG-16 model demonstrated superior performance, followed by VGG-19 as compared to the other models. As well, tests for statistical significance showed that the VGG-16 model demonstrated a tighter CI and a smaller margin of error for the AUC values as compared to the other models. This may be because the architectural depth of VGG models is optimal for the current task as compared to the other DL models used in this study.

We observed that the weighted averaging ensemble outperformed the other ensemble methods. We believe this may be due to the fact that deeper models are not always optimal for all the tasks, especially in the medical domain, where the data distribution is different compared to stock photographic images. We empirically observed that the VGG-16 model demonstrated superior performance in all performance metrics, followed by VGG-19, as compared to the other models. By awarding higher weights to the predictions of the best performing VGG models, we were able to improve the performance as compared to the other ensemble methods.

We chose CRM as our visualization method for localizing abnormal CXR regions based on the recent study (*Kim, Rajaraman & Antani, 2019*) where CRM shows a better localization performance than other existing methods including CAM and Grad-CAM. All individual CNN models employed in our study are found to have a different classification and localization error distributions. Therefore such errors could be compensated or reduced by our ensemble approach that combines and averages the individual localization performance. Our experimental results also show that our ensemble models significantly outperform the best-performing individual CNN model.

To summarize, modality-specific transfer learning helped to improve performance and generalization in a related target task. The performance of the ensemble improved with using the models that inherited modality-specific knowledge from a large-scale CXR data collection. The performance of ensemble visualization improved with the use of models that benefited from modality-specific knowledge transfer and provided a combined prediction that is superior compared to any individual constituent model.

## CONCLUSION

The combination of modality-specific model training and ensemble learning helped to: (a) transfer modality-specific knowledge that is repurposed to improve classification performance in a similar task; (b) reduce prediction variance, sensitivity to the training data, and model overfitting; and, (c) improve overall performance, and generalization. However, ensemble methods are computationally expensive, adding training time and memory constraints to the problem. It may not be practicable to implement model ensembles at the present; however, with the advent of low-cost GPU technology and availability of high-performance computing solutions, model ensembles could become practically feasible for real-time applications. Ensemble visualization helped to interpret the models' behavior, compensated for the error of missing ROIs using individual CNN models and demonstrated superior ROI detection and localization performance as compared to any individual constituent model. Further, CRM offers improved interpretation and understanding of the model's learned-behavior. We believe that the results proposed are valuable toward developing robust models for medical image classification and ROI localization. Future studies could explore the application of ensemble CRMs to other diagnostic/screening applications, for example, detecting cancers in various image modalities, separating pneumonia from TB in pediatric CXRs, skeletal malocclusion detection from 3D-Cone Beam CT images, etc. In addition, it is desirable to compare the performance of ensemble CRMs to other state-of-the-art visualization strategies.

### Funding

This work was supported by the Intramural Research Program of the National Library of Medicine (NLM), National Institutes of Health (NIH) and the Lister Hill National Center for Biomedical Communications (LHNCBC). The intramural research scientists (authors) at the NIH dictated study design, data collection/analysis, decision to publish and preparation of the manuscript.

### Grant Disclosures

The following grant information was disclosed by the authors:
Intramural Research Program of the National Library of Medicine (NLM).
National Institutes of Health (NIH).
Lister Hill National Center for Biomedical Communications (LHNCBC).

### Competing Interests

The authors declare there are no competing interests.

### Author Contributions

- Sivaramakrishnan Rajaraman conceived and designed the experiments, performed the experiments, analyzed the data, prepared figures and/or tables, authored or reviewed drafts of the paper, and approved the final draft.
- Incheol Kim conceived and designed the experiments, performed the experiments, prepared figures and/or tables, authored or reviewed drafts of the paper, and approved the final draft.
- Sameer K. Antani conceived and designed the experiments, analyzed the data, authored or reviewed drafts of the paper, and approved the final draft.

### Data Availability

Codes are available at: https://github.com/sivaramakrishnan-rajaraman/Detection-and-visualization-of-abnormality-in-chest-radiographs-using-modality-specific-CNNs.

ChexPert dataset is available upon completion of the CheXpert Dataset Research Use Agreement at: https://stanfordmlgroup.github.io/competitions/chexpert/.

Kaggle CXR dataset is available at https://www.kaggle.com/c/rsna-pneumonia-detection-challenge/data.

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
