# Peer review of "Detection and visualization of abnormality in chest radiographs using modality-specific convolutional neural network ensembles"

_PeerJ, doi:10.7717/peerj.8693_

## Round 0.1 · original submission · Major Revisions

While the reviewers are generally positive about the manuscript, a significant number of questions are apparent in their reviews. Please address the reviewer concerns with particular emphasis on improving the overall clarity and explanation of the methods and results.

Reviewer 1 ·

Basic reporting

Although the complexity of the content to solve a hard problem, the manuscript is clearly written in a professional, unambiguous language. It is well structured and easy to read.

The literature employed is pertinent and actual. But is suggested to authors include and consider some current works in this research:
- Efficient Deep Network Architectures for Fast Chest X-Ray Tuberculosis Screening and Visualization (https://doi.org/10.1038/s41598-019-42557-4);
- Deep learning in chest radiography: Detection of findings and presence of change (https://doi.org/10.1371/journal.pone.0204155);
- Deep learning for chest radiograph diagnosis: A retrospective comparison of the CheXNeXt algorithm to practicing radiologists (https://doi.org/10.1371/journal.pmed.1002686);
- Computer-aided detection in chest radiography based on artificial intelligence: a survey (https://doi.org/10.1186/s12938-018-0544-y).

The figures are relevant and present good quality. It is suggested a better explanation of figure 4, to show the source from base models with ImageNet; figure 6, for meta-learner should be provided more details; figure 9, what means the Thr: 0.1? And in figures 10 and 11, should be explained the impact of a satisfactory precision penalizing the recall metric. Figure 5 could be removed.

Experimental design

It is perceived that the core of work is the modality-specific learning. Like observed, that is a refined training under the classical CNN based on ImageNet (lines 268-9 and 449), with a wide CXR dataset (contains 18 different abnormalities) and refined again with a specific dataset (RNSA – specific for pneumonia). This process was namely of coarse-to-fine learning.

I consider that the authors use another nomenclature, modality-specific learning, to refer to the classical and commonly used techniques of transfer learning. If that is different, I suggest the authors differentiate it from transfer learning. If the same, the authors should adopt the term transfer learning to refer to this approach.

Attention for lines 174-6, for improving the explanation, it is understood that modality-specific is applied under the CheXpert dataset, but the successive repurpose for detecting abnormality in the RSNA dataset is a kind of modality-specific (or transfer learning) under a specific abnormality (pneumonia). The question about localizing regions is a function of visualizing techniques and doesn’t be the responsibility of methods for detecting abnormalities.

Contrarily, like exposed in line 180, the work doesn’t propose the visualization techniques. The visualization technique was proposed in another study (https://doi.org/10.3390/diagnostics9020038) from the same authors.

Opposed with the description in lines 182-5, it is not the first time that this kind of approach was realized (please see https://doi.org/10.1038/s41598-019-42557-4). This contribution should be reconsidered.

In relation of the main objective, in identify abnormalities or not in CXR, is perceived the use of widely imbalanced dataset (~1:10 for CheXpert and 1:3 for RSNA), but the authors don’t mention this question and its impact in the evaluation.

Sometimes is confused and hard to understand the work, that tries to aggregate two complex tasks, detecting and visualizing (like a way of interpreting). Considering the ambitious task to solve both problems conducted the work to lose quality. It is a little confusing, the proposal for detecting under several abnormalities from a dataset and just one for another dataset, and the visualization focused under the pneumonia dataset. The authors should reconsider what is the real contribution, delimit the scope and focus the work in your main question.

The authors should specify the difference of the dropout U-Net and the "all dropout" (like specified in Fully Convolutional Architectures for Multi-Class Segmentation in Chest Radiographs - arXiv:1701.08816). Should be explained in more detail why this dropout ratio in 0.2 impacts more than the original research that specifies this ratio in 0.1?

What means the Custom model (table 4)? Is it the same of the Baseline (tables 2 and 3)? And Pretrained CNNs are based too on the weights from ImageNet? This confused question should be better explained in the manuscript.

Why use CRM as a unique way of model visualization?
Why not consider an evaluation with other techniques, like Saliency Maps ((arXiv:1312.6034v2) widely mentioned in the current literature). For example, the recent work from Scientific Reports (https://doi.org/10.1038/s41598-019-42557-4) reports the success of this method for visualizing and localizing Tuberculosis in CXR.

In ensemble CRM is not clear how the CNN models complement each other if there is some disagreement with the different models? How the aggregation of this supposed discordance of the individual architectures to compose a final suggestion works? For example, if there is a more precise CNN model, why I have to believe in a CNN model that disagrees with this decision? It is suggested to authors provide more elements and better explanations to justify this question.

Validity of the findings

It is perceived the effort of authors in provided an extensive and challenging task, but some questions should be clarified and better detailed to highlight the real contribution of this work in function of its focus.

In a general view, is unclear the real contribution and innovation of the work. The contribution of the weighted averaging ensemble comes from (https://doi.org/10.1109/embc.2019.8856715) not cited; the visualization and localizing come from (https://doi.org/10.3390/diagnostics9020038). Then, the core of the work is the modality-specific learning, that I consider the classical use of transfer learning (with the use of baseline and pre-trained CNNs models under ImageNet) with another name.

As pointed by authors there are at least 18 abnormalities in CheXpert. RSNA treats only pneumonia, 1/18 abnormalities of CheXpert. The authors don’t mention anything about that, e.g., the diagnosis of pneumonia is easily confused with tuberculosis. And not is clear, what was the real influence of RSNA dataset, with a specific abnormality, when aggregated to fine learning. Like exposed in lines 317-20, the best ensemble was the weighted averaging, based on RSNA. It is suspect that the work could be a bias to pneumonia against a general CXR identification.

The description of CheXpert should specify how many instances there are for each abnormality.

In table 3, how exactly are determined the best layers? The authors should answer this question in more details

Like exposed, lines 446-50, with the explanation of table 4. Not is clear the evaluation process. Were all the evaluations, in terms of detection of abnormalities or not, done and reported considering the reserved test data from both datasets?

The problem that difficult a better understanding is the use of 2 different datasets for predicting abnormalities. It is recommended that the authors supply the results for each individual dataset. And for CheXpert dataset shows the results for each abnormality.

It is not clear the methodological process applied for evaluating the ROI of abnormal images (RSNA pneumonia). The figures 10 and 12 demonstrated didactics aspects, but should be specified if this evaluation was done under the 17833 abnormal images from the dataset. And what was the success rate obtained from this set of images?

The use and efficiency of metrics IoU and mAP don't are clear. There aren't academic works that reported the use of them. I consider it very dangerous to rely on these metrics. The authors should justify its adoption in better detail. Besides that, the authors don’t explain the expected values for these metrics (what is the range?) and what is the impact of the difference presented between tables 6 and 7.

The IoU rate, presented in figures 9 and 12, doesn't correspond with the rates presented in tables 6 and 7. What justifies these differences? Why just this metric is presented, and mAP?

I believe the authors could improve the conclusion; the work seems a lot of questions to evolve. However, the authors don’t mention any future work.

Additional comments

General typos:
- line 154: Huang G in the citation in place of just Huang;
- line 255: CheXpert in place of Chexpert;
- line 445: grid in place of gird;
- Define a pattern for IoU (or IOU) in text and in figures 10 and 11.

Reviewer 2 ·

Basic reporting

The authors describe a multistep method to train an ensemble of CNNs to classify chest x-rays as normal/abnormal and identify the region of abnormality.

They train a U-Net to generate lung masks (what role did these play?).

They train a variety of different CNN architectures to a training partition of the CheXpert dataset and report 'coarse model' (i.e. single CNN architecture) performance on this dataset.

These base learners are then used as the pre-trained models in a second step to fine-tune them again on the RSNA pneumonia dataset. After this process, certain near-final convolutional layers from these base learners are then chosen (through a process that is not clear to me - Figure 4 / text do not provide sufficient detail - line 294 - “We instantiated the convolutional part of the coarse models and extracted the features from different intermediate layers with an aim to identify the optimal layer for feature extraction toward the current task.”).

These extracted layers were then concatenated and another stacked model was trained for classification on RSNA CXR data. Additional hand-constructed ensembles of the model predictions were also created.

Ensemble Class-selected Relevance Mapping is used to demonstrate how ensembling can be used to improve various accuracy figures of abnormality detection as measured by IoU/mAP.

The authors report their process improves various normal/abnormal classification metrics (AUC, etc) and the accuracy per IoU/mAP of regions of abnormality. They conclude that their multiple-step training process and ensembling led to improved performance.

The literature review is good and much relevant work is discussed. The authors performed extensive experiments and report their results completely, although I believe that important experimental details are not fully explained and there are key parts of the analysis that are unclear to me (see below).

Experimental design

I summarized my understanding of the approach above.

There is room to explore how ensembling can improve predictions in medical imaging, and there has been fairly limited exploration of this so far, so the topic is important. My questions about the experimental design follow:

1. In the first step to train coarse learners - are you training to all CheXpert classes or to normal/abnormal target and then inferring abnormality? This should be described.

2. In the second step to train fine-tuned learners - normal/abnormal here just means pneumonia/no pneumonia, since this is how the data is labeled? This should be described.

3. It is not obvious to me that training on the second dataset improved performance. In Table 4, AUCs on coarse models are presumably evaluated on CheXpert, and AUCs on fine models are evaluated on RSNA data? In which case they're not comparable. If they're both evaluated on RSNA data, this should be described. Even then, it wouldn't be surprising that this improved performance - more data, and specific to the dataset. The more interesting question would be if performance on the original CheXpert dataset was better or worse after seeing data from a different modality.

4. In Table 5, ensembling performance is reported, presumably on RSNA test data (needs to be stated). A statistical test should be included to demonstrate that the AUC improvement to 0.97 is actually significant over the fine-tuned models - it's not obvious to me that it is.

5. In the stacking part of the ensembling experiment - were these combinations also trained on the RSNA train data? It is important to make clear they were not trained on test data. It is somewhat surprising that stacking classification performance was inferior to simple averaging - why do the authors believe this was?

6. "Table 3 lists the best-performing empirically determined model layers. The performance metrics achieved through extracting the features from the optimal layers of the different coarse CNN models toward classifying normal and abnormal CXRs are shown in Table 4." --> how were these layers specifically chosen? The text is not clear on this: "We instantiated the convolutional part of the coarse models and extracted the features from different intermediate layers with an aim to identify the optimal layer for feature extraction toward the current task." This needs to be clearly and fully described - Figure 4 is far too superficial given the complexity of the experiments. Why was it necessary to extract the optimal layer for classification if a classification model was trained?

7. Table 7- how are these region of interest ensembles created? The text says they are generated from the top performing "CNN models that are selected based on the IoU and mAP scores as shown in Table 5" - but these scores / models are not identified in Table 5, which discusses ensemble classification performance. Were they constructed only using RSNA training data, or were you allowed to optimize to test data in constructing these models? Obviously there will be a big bias towards improved results if you were able to use test data in constructing them. Improved explanation of how these ensembles were created is needed, and provided test data was not used in their construction, statistical testing should be included to demonstrate that the performance improvement was significant (or to conclude that it wasn't, if not).

8. “We removed the mapping score value below 10% of the maximum score in each CRM to minimize a possible noisy influence of a very low mapping score during the ensemble process.” - I do not understand this, can the authors explain?

9. What role did the lung masks serve in this work? Were all the images cropped out to exclude what wasn't in the lung mask in training?

10. Mention should be made of what dataset is being used for evaluation in all tables (CheXpert test, RSNA test, etc.)

Validity of the findings

11. Overall - I appreciate the extensive experiments the authors performed and the thoughtful content that is included in this work. However, the number and variety of different experiments run ends up being a hindrance to the final work, as the paper ends up disjointed and very difficult to follow. Multiple steps are performed sequentially and evaluated on different datasets - lung masking, then training on one dataset, then fine tuning on another, then extracting certain layers, then ensembling, and both classification and segmentation accuracy are assessed - and it is hard to draw any firm conclusions about how ensembling contributes to the overall stack given how many moving pieces there are. These clarifications are needed because despite the work being based on open-source datasets, insufficient details are given to allow authors to replicate the work.

12. Many of the technical details could be moved to a supplement, and more attention could be given to clearly outlining the various steps of the experiment and how they fit together, both in text and in figures as possible. The datasets used (and the partitions) need to be much more clearly stated for each step, and the reader needs to be assured that test was not used in the ensembling process except for final evaluation. Figures need to be used more effectively to display the complexity of the experiments. As much as possible, the contribution of ensembling needs to be isolated by providing apples to apples statistical testing against non-ensembles models on the same dataset.

13. Finally - the authors could alternatively simplify the experiment. If they wish to isolate the effect of ensembling on predictive accuracy, they could use their original results from CheXpert to develop an ensemble of models, evaluate it on CheXpert test, and (if they desired) show how this ensemble generalizes to RSNA as an additional validation. If they wish to focus on how ensembling improves the region of abnormality detection, they could limit their experiments to this on RSNA data. If the authors feels strongly that their full pipeline is needed, the various components need to be individually defended and justified.

I appreciate having had the chance to review this research and thank the authors for their contribution.

---

## Round 0.2 · Major Revisions

Thank you for addressing the reviewer concerns. There are, however, a few issues that remain to be addressed before acceptance. In particular, some aspects of the experimental design as brought up by Reviewer 2 are unclear along with some potential clarifications needed on some aspects of the work as brought up by Reviewer 1.

Reviewer 1 ·

Basic reporting

It is perceived in this revised version of the manuscript that the authors attend the major question demanded in the evaluation of the first version. The authors provided a more clear justification for the main questions and salient the contributions of the work.

Then I consider that some minor questions (listed below) should be attended to allow the acceptance of the work:

a. About class weight for the imbalance issue, the authors should cite at least one strong study to prove the efficiency of this kind of approach, I saw it specification only in practical issues and I consider it a little dangerous to rely just on the implementations without a deeper investigation if there are other recognized approaches to treat the imbalanced data. This issue is very pertinent for the success attained by the work and for the proposed approach;

b. In relation to this question: “Q12. In ensemble CRM is not clear how the CNN models complement each other if there is some disagreement with the different models? How the aggregation of this supposed discordance of the individual architectures to compose a final suggestion works? For example, if there is a more precise CNN model, why I have to believe in a CNN model that disagrees with this decision? It is suggested to authors provide more elements and better explanations to justify this question.”, I do not see an explicit manifestation of the authors in the text to provide improvements in relation to this question;

c. I consider the questions: “Q18. It is not clear the methodological process applied for evaluating the ROI of abnormal images (RSNA pneumonia). The figures 10 and 12 (in the first version of the manuscript) demonstrated didactics aspects but should be specified if this evaluation was done under the 17833 abnormal images from the dataset. And what was the success rate obtained from this set of images?” and “Q20. …What justifies these differences? Why just this metric is presented, and mAP?” until unclear and I believe that the authors could explain in more detail to clarify this question. I understand that the reported values for IoU and mAP are the maximum reached, but how representative are the values reported in tables 7 and 8 in terms of the expected range of them? For example, if the range of them is the interval [0.1 0.6], the 0.383 for IoU or 0.377 for mAP to VGG-16 reported in table 7 or a 0.433 for IoU or 0.477 for mAP to ensemble-5 in table 8, not seems very good values. And, in figure 11, the example provided for individual or ensemble CRM not reflect the values reported in tables 7 and 8. How was reached these superior values of IoU presented in the example (sometimes superior to the range 0.6) in figure 11 for individual and ensemble CRM?

d. I suggest an improvement in the conclusion section, I see several evidences that conduct for extension of this work in future works, like for example a comparative with salient maps method for visualizing, or the extension of another test data set not related to only an abnormality, like pneumonia. But the authors did not mention any future strategy.

Experimental design

no comment

Validity of the findings

no comment

Additional comments

no comment

Reviewer 2 ·

Basic reporting

1. I appreciate the continued efforts the authors have made to improve this work. The clarity of the methods used in the manuscript are substantially improved.

The additional captions on the tables clarified a number of my questions. The improved figures, especially those highlighting how ensemble CRMs can improve performance, are helpful.

The suggestions provided by Reviewer 1 and implemented by the authors have further strengthened the literature review.

Experimental design

2. Re: significance testing. I did not understand the authors' reply about cross-validation. I am not suggesting that cross-validation be performed. Significance testing for AUCs can be performed with the predictions they have already generated on a fixed test set. For an example, see Gulshan et al providing confidence intervals for an ROC curve:

https://jamanetwork.com/journals/jama/fullarticle/2588763

and see testing available in in

https://cran.r-project.org/web/packages/pROC/pROC.pdf

In the machine learning literature, it is unfortunately standard to report results without any statistical testing. Given that this work is oriented towards the medical/biological community, I believe significance testing -- at least of the core claims asserted in the conclusion -- is needed. It is not obvious to me that an experimental AUC of 0.97 (ensembling classifier performance on Chexpert test) is truly superior to an AUC of 0.96 (VGG-16 performance on same), but it is claimed that ensembling improved classification performance in the paper. There should be a statistic to back up each of these claims made in the conclusion:

"The combination of modality-specific model training and ensemble learning helped to: a) transfer modality-specific knowledge that is repurposed to improve classification performance in a similar task; b) reduce prediction variance, sensitivity to the training data, and model overfitting; and, c) improve overall performance and generalization. The proposed strategy shows improved performance and generalization as compared to other non-ensemble methods."

3. As excerpted above, the authors assert that both modality-specific model training and ensemble learning offered a range of benefits. I believe they now show that ensembling offers benefit for generating more reliable bounding boxes and possibly classification performance on their data, but I still do not find their claims credible that modality-specific model training offers any benefit.

The authors are fine-tuning a classification model twice -- once on Chexpert data, then separately on RSNA data. The results in table 5 do *not* support that there is a benefit from this procedure. They compare 'baseline' model performance (ie, ImageNet-initialized, then fine-tuned on Chexpert data only) to 'fine' model performance (i.e., ImageNet-initiailzed, then first fine-tuned on Chexpert data, then fine-tuned on RSNA data).

Their results are consistent with there being a benefit from fine-tuning on RSNA data. This is not surprising, since the classification task is different. Normal/abnormal classifications based on Chexpert evaluates for a wider set of pathology than RSNA, where normal/abnormal only evaluates for pneumonia. It is entirely expected that fine-tuning on more accurate labels would improve performance. (This fact needs to be addressed by the authors in the Discussion.) This does not demonstrate any benefit from first fine-tuning on Chexpert data.

If the authors wish to claim that the modality-specific model training pipeline of fine-tuning twice on separate datasets with different labels offers value, they would need to include a new category in Table 5 for each type of CNN for a model that was initialized to ImageNet pretrained weights and then only fine-tuned on RSNA data. That could demonstrate that initializing to the Chexpert fine-tuned weights helped them learn a better solution than initializing to ImageNet only before fine-tuning. Otherwise, this claim is not defensible. This is a core claim of the paper, so it is very important that it either be convincingly made or removed.

It would also be helpful to use the same terminology of 'coarse' instead of 'baseline' in Table 5 so that 'coarse' and 'fine' can be contrasted.

4. Related to the above - since the model is again fine-tuned on RSNA data, comments about generalization ("improved generalization in transferring knowledge") should not be included, as there is no real test of generalization being performed. Only if there were a new test dataset that was not used for training could we really assess generalization.

Validity of the findings

5. Overall, I think the strongest part of this work as it stands is demonstrating that ensembling can be useful for creating better CXR saliency maps, and possibly classification performance if statistical testing bears it out.

I believe that this contribution is obscured by the complexity of the pipeline created to facilitate modality-specific model training. My primary remaining concern is that claims about the benefit of the modality-specific training be either substantiated or removed, as discussed above.

---

## Round 0.3 · accepted · Accept

Thank you for addressing the reviewer concerns. Congratulations again!

Reviewer 1 ·

Basic reporting

I consider the authors provided satisfactory answers to my questions and clarify some points obscured in the previous version.

The question about class weight was attended with the addition of pertinent literature.

In relation to ensemble CRM was provided a better explanation that helps the reader to understand the superiority of the ensemble strategy.

Although I consider 20% to test few samples to be evaluated, the authors provided a better explanation for the readers.

Additionally, the authors follow the suggestion from Reviewer 2 with the inclusion of confidence intervals.

In relation to the rates for IoU and mAP, was clarified the determination from the precision-recall curves. However, I consider that this explanation "We agree to the fact that some images have the IoU scores superior to the range 0.6 because this score quantitatively indicates the degree of overlap between the bounding box surrounding the detected ROI and the ground-truth bounding box" should be incorporated into the manuscript to avoid miss interpretations.

Experimental design

no comment

Validity of the findings

no comment

Reviewer 2 ·

Basic reporting

no comment

Experimental design

no comment

Validity of the findings

no comment

Additional comments

Thanks again to the authors for their detailed and thoughtful revisions of their manuscript. These revisions constructively address my previous concerns, and I recommend acceptance of this manuscript in its current form.